Optimized models and deep learning methods for drug response prediction in cancer treatments: a review

http://orcid.org/0000-0003-3011-9072 Hajim Wesam Ibrahim 1 2 p106883@siswa.ukm.edu.my
Zainudin Suhaila 2
Mohd Daud Kauthar 2
Alheeti Khattab 3
1 Department of Applied Geology, College of Sciences, Tirkit University , Tikrit, Salah ad Din , Iraq
2 Center for Artificial Intelligence Technology, Faculty of Information Science and Technology, Universiti Kebangsaan Malaysia , Selangor , Malaysia
3 Department of Computer Networking Systems, College of Computer Sciences and Information Technology, University of Anbar , Al Anbar, Ramadi , Iraq
Alatas Bilal
Electronic publication date: 2024 Mar 25
Publication date: 2024
Volume: 10
Electronic Location ID: e1903
Received 2023 Sep 5; Accepted 2024 Jan 31
Copyright: © 2024 Hajim et al.
Copyright year: 2024
Copyright holder: Hajim et al.
License: This is an open access article distributed under the terms of the Creative Commons Attribution License, which permits unrestricted use, distribution, reproduction and adaptation in any medium and for any purpose provided that it is properly attributed. For attribution, the original author(s), title, publication source (PeerJ Computer Science) and either DOI or URL of the article must be cited.
License URL: https://creativecommons.org/licenses/by/4.0/

Keywords: Deep learning, Machine learning, Drug response prediction, Cancer diagnostic, Precision medicine

Funding: Centre for Artificial Intelligence Technology Faculty of Information Science & Technology Universiti Kebangsaan Malaysia Ministry of Higher Education Ministry of Higher Education and Fundamental Research FRGS/1/2022/ICT02/UKM/02/7 Research University Grant GUP-2020-089 This research was supported by the Centre for Artificial Intelligence Technology, Faculty of Information Science & Technology, Universiti Kebangsaan Malaysia and the Ministry of Higher Education and Fundamental Research through grant code [FRGS/1/2022/ICT02/UKM/02/7] and through the Research University Grant with the code (GUP-2020-089). The funders had no role in study design, data collection and analysis, decision to publish, or preparation of the manuscript.

==============================
Recent advancements in deep learning (DL) have played a crucial role in aiding experts to develop personalized healthcare services, particularly in drug response prediction (DRP) for cancer patients. The DL’s techniques contribution to this field is significant, and they have proven indispensable in the medical field. This review aims to analyze the diverse effectiveness of various DL models in making these predictions, drawing on research published from 2017 to 2023. We utilized the VOS-Viewer 1.6.18 software to create a word cloud from the titles and abstracts of the selected studies. This study offers insights into the focus areas within DL models used for drug response. The word cloud revealed a strong link between certain keywords and grouped themes, highlighting terms such as deep learning, machine learning, precision medicine, precision oncology, drug response prediction, and personalized medicine. In order to achieve an advance in DRP using DL, the researchers need to work on enhancing the models’ generalizability and interoperability. It is also crucial to develop models that not only accurately represent various architectures but also simplify these architectures, balancing the complexity with the predictive capabilities. In the future, researchers should try to combine methods that make DL models easier to understand; this will make DRP reviews more open and help doctors trust the decisions made by DL models in cancer DRP.

Introduction

Cancer precision medicine is an effective strategy to fight cancers by bridging genomics and drug discovery to provide specific treatment for patients with different genetic characteristics (Liu et al., 2023). Precision medicine, commonly called customized medicine, has arisen as an avant-garde idea and poses a considerable challenge for the twenty-first century. Then the genomics-based concept of precision medicine began to emerge following the completion of the Human Genome Project. In contrast to evidence-based medicine, precision medicine will allow doctors and scientists to tailor the treatment of different subpopulations of patients who differ in their susceptibility to specific diseases or responsiveness to specific therapies (Wang & Wang, 2023). A statistical approach to precision medicine uses patient-level data and formalizes clinical decision-making into a model composed of patients’ features, key decision points, treatment options at those decision points, and primary outcomes to be optimized. Rather than relying on known or hypothesized relationships between interventions, health status, and outcomes, statistical and machine learning analytics can learn directly from the data how to tailor treatments optimally (Kahkoska, Freeman & Hassmiller Lich, 2022). The extensive use of technical resources has considerably facilitated this approach, allowing for the analysis, treatment, and building of models to aid in cancer detection and treatment (Chang et al., 2018; D’Orazio et al., 2022; Tansey et al., 2022). Using in silico models (Kather et al., 2018; Hadjicharalambous, Wijeratne & Vavourakis, 2021) has sped up research a great deal, making it possible to quickly create models that can be changed to fit the needs of each patient (Vougas et al., 2019). Furthermore, the availability of multi-omics data has aided the practicality and widespread adoption of precision medicine in cancer treatment. These extensive datasets provide a thorough grasp of the distinct properties of each cancer sample. Details on gene expression, copy number variation, genetic mutation, methylation patterns, and proteomics information are all included in these massive multi-omics datasets (Goldman et al., 2018; Zhang, Chen & Li, 2021). In the field of precision medicine for anticancer treatment, accurate prediction of drug response is crucial (Ali & Aittokallio, 2019; Wu et al., 2020; Güvenç Paltun, Mamitsuka & Kaski, 2021; Zuo et al., 2021). The main objective is to understand the underlying molecular processes that drive tumor growth and utilize appropriate biomarkers to customize medications for specific molecular subgroups to target malignancies precisely. This approach emphasizes personalized treatment for each patient, considering their distinct molecular, immunological, and other biological traits (Amin, Chiam & Varathan, 2019; Tsimberidou et al., 2020; Bien et al., 2021).

As a result, drug response prediction (DRP) is critical in various disciplines. New research articles in this domain emerge regularly, harnessing learning algorithms for DRP. In order to effectively manage the increasing volume of publications, a recent issue of Briefings in Bioinformatics was singularly dedicated to the investigation of DRP in cancer models (Ballester et al., 2022). Statistical and machine learning (ML) algorithms have recently been used as alternative models to assess the sensitivity of anticancer medicines (Chen & Zhang, 2021). Given the amount of drug screening data, it is critical to prioritize computational technique research, development, and application strategies. This method is vital for gaining trustworthy insights into the development of medicine. The existing body of literature has already documented a diverse set of methodologies and computer models used in DRP research, resulting in the identification of drug response biomarkers. Numerous ML-based techniques have been employed in these approaches (Adam et al., 2020). Neural network models (Xia et al., 2021) and Bayesian multitask multiple kernel learning (Bernard et al., 2017; Manica et al., 2019) are some of these methods. Others are random forests (RF) (Gayvert et al., 2017), support vector machines (SVMs) (Huang et al., 2017), and naïve Bayes (Kang et al., 2018; Anagaw & Chang, 2019; Patel et al., 2020). Furthermore, recent research has increasingly focused on utilizing ML algorithms to address these issues.

Machine learning (ML) methodologies have found extensive application in drug response prediction (DRP) through the analysis of clinical screening data. Nevertheless, these ML algorithms are prone to encountering false positives, and their diminished accuracy has led to a discernible reduction in their utilization in recent years, as highlighted by Xia et al. (2021) and Su et al. (2022). Furthermore, it is noteworthy that while ML algorithms can provide elevated prediction accuracy, their performance is weak when using large-scale data; they consume more extended training and testing times due to their inherent complexity (Adam et al., 2020).

The deep learning (DL) methods are more efficient than the traditional ML algorithms for predicting single-drug effectiveness and the challenging multi-drug approach for anticancer treatments (Chen et al., 2022). In addition, DL is an alternative tool for learning the internal and low-dimensional representation of the data collected from pre-screening data (Tanebe & Ishida, 2021). On the other hand, DL algorithms have exhibited accurate findings and excellent performance in various sectors, including DRP, due to their capacity to learn complicated features and eliminate false positives.

But, even though computational models and medical experts are working together, more research needs to be done on how to improve the predictive power, generalizability, and interpretability of computational models that have already been proposed.

With their shallow architecture, ML algorithms often struggle to capture all the intricacies of cancer and drug data (Daoud & Mayo, 2019; Lundervold & Lundervold, 2019). When looking at huge medical datasets that include gene expression, medicines, omics, cell lines, and protein-to-protein interactions, ML algorithms need a lot of time to train and become more complicated in terms of time and space.

On the other hand, DL algorithms are better suited for large cancer datasets because they can comprehend complicated disease traits and quickly adapt to large-scale datasets (Chen et al., 2022). Although they have the same connectionist structure as neural networks, deep neural networks (DNN), recurrent neural networks (RNNs), and convolutional neural networks (CNN) for supervised learning are considered distinct classes of ML algorithms based on artificial neural networks (NNs); additionally, autoencoders are used for unsupervised learning. The DL was a predictive model in various activities linked to cancer medication development (Tranchevent, Azuaje & Rajapakse, 2019). The most typical deep learning architectures are shown in Fig. 1.

Figure 1 The most typical deep learning architectures.

(A) A deep neural network. (B) Convolution neural network. (C) Recurrent neural network. (D) Auto encoder.

DL also does a better job of predicting how drugs will work than other ML models because it is more complex in its connectionist form and can be built in a variety of mathematical ways (Tanebe & Ishida, 2021). As such, DL-based models have gotten much interest in recent years for their use in DRP. A comprehensive review undertaken by Baptista, Ferreira & Rocha (2021) thoroughly reviews the available works on DL-based approaches in the field of DRP. This study aimed to investigate the current progress in DL-based DRP algorithms. The focus was on summarizing existing methodologies and evaluating their approaches and performance. The architectures covered in this study included the CNN, RNN, and deep self-coding networks (DAENs).

Additionally, this study encompassed various methodologies beyond supervised and unsupervised learning. Notably, it compiled a collection of the most representative studies that built the frequently used DL models in these applications (Anwar Lashari et al., 2018; Abdelhafiz et al., 2019; Chriskos et al., 2021; Cole et al., 2021). The evaluation primarily assessed the strengths and weaknesses of the study to draw potential research conclusions. This review aims to enhance understanding of the operational principles of DL-based DRP approaches and explore possible future directions in this domain.

Survey methodology

The current study indicates that optimal deep-learning technologies are used in modern medical science. This research also examines the possibilities of using optimization methodologies to take a practical and advanced step toward increasing the accuracy of forecasting medication responses in deep learning.

While in machine learning, some efforts have been made in the literature to optimize the model for drug response prediction (Gao et al., 2020; Mohammadi-Balani et al., 2021; Nguyen et al., 2022). On the other hand, research into optimizing deep learning models still needs to be completed.

A full review of the literature using certain keywords was done to look at research that specifically improves deep learning models for predicting how medications will work. The first step is to identify keywords that will be used to search online literature databases for peer-reviewed articles. The grouping chosen was (deep learning, drug response, and cancer). We then searched the PubMed scientific databases for studies with these phrases in their titles, abstracts, or keyword lists.

Recently, there has been a significant increase in studies utilizing DL algorithms for DRP, emphasizing pre-processing, feature learning, dimensionality reduction, and data classification.

Mohamed, Zainudin & Ali Othman (2017) implemented a metaheuristic-based filter approach to select relevant genes for drug response microarray classification. Three evolutionary algorithms were utilized to filter the top genes in high-dimensional drug microarray datasets. Subsequently, three machine learning models were applied for drug response classification. The metaheuristic approach is better than traditional filters at picking out the most important features. It achieved 71.43% accuracy for tumors in the central nervous system, 87.10% accuracy for tumors in the colon, and 100% accuracy for tumors in the lungs.

Zhang et al. (2018) introduced a heterogeneous network-based method for DRP called HNMDRP. This model incorporates similarity measures to construct a network that considers the heterogeneous characteristics of the drug, cell line, and target relationships. Using a data-driven algorithm, the proposed model figures out model-based profiles. This leads to better performance measures (F-score) than current methods, with AUC improvements of 2% to 25%. However, the method is limited because it only uses drug, target, and cell line similarity networks without taking into account cell line variation, multidimensional features, or relationship information. This means that the accuracy of the predictions is only moderate.

Preuer et al. (2018) introduced DeepSynergy, a deep neural network-based model for DRP. It incorporates compound and genomic information as inputs and utilizes a normalization strategy to handle data heterogeneities. The model includes canonical layers to capture synergies between different drugs, enabling the detection of specific combinations with maximum efficacy. DeepSynergy improved performance by 7.2% by making a combined representation of drug and cell lines that work well together. They reported a Pearson correlation of 0.73 and an AUC of 0.90. However, the model’s generalization ability may need to be improved when dealing with fewer drugs and cell lines.

Chang et al. (2018) proposed CDRscan, a deep learning-based model for determining cancer drug response profiles, in a separate study. This model employs a two-step convolution architecture. It demonstrated high prediction accuracy, with an R2 value of 0.84 and an AUROC value of 0.98. There was less variation in the data, though, so the predicted accuracy for a few compounds from the genomics of drug sensitivity in cancer (GDSC) datasets was lower. Although the model performs well, data availability can impact the accuracy of predictions for specific compounds in the dataset.

Chen et al. (2018) devised a DRP model that combines deep belief networks (DBN) and ontology fingerprints to generate ontological gene profiles. This model exhibited high performance, achieving an F-measure of 92% even in scenarios with imbalanced data. However, the model’s training capability is limited.

Matlock et al. (2018) developed a model stacking approach for DRP utilizing the random forests (RF) algorithm within a deep learning architecture. This method automatically reduces the inherent bias in RF-based models. The proposed approach achieved a high AUC of 0.9 and reduced errors, with eigenvalues of 0.95 and 0.23. However, the model only addresses linear bias in stacking and does not handle nonlinear bias.

Xia et al. (2018) introduced a DRP model employing deep residual neural networks (ReNN) with two objectives: encoding treatment features and predicting tumour growth. This model effectively captures the combination of molecular characteristics (gene expression, microRNA, and proteome), resulting in an increased response variance of 94%. The model achieved a Pearson correlation of 0.972, a Spearman correlation of 0.965, and a coefficient of determination (R2) of 0.94. However, it slightly underperforms compared to the hyperparameter-optimized model.

Tan et al. (2019) developed a DRP model utilizing ensemble learning and incorporating drug screen data and two novel signatures. An elastic net, pairwise SVM, kernelized Bayesian multitask learning, and multitask neural networks are some of the tools that are used in this method. The proposed solution attained mean square error (MSE) values of 2.03 and 4.496 for the GDSC and cancer cell encyclopedia (CCLE) datasets, respectively. However, this method does not consider cancer relationships among genes from the sub-networks.

Chiu et al. (2019) introduced a DRP model that combines genomic profile information with a deep neural network (DNN). Transfer learning is employed, where a pre-trained neural network in a large pan-cancer dataset serves as an encoder integrated into the DRP network. This model pipeline extracts features from cancer cells and drug sensitivity, achieving a mean square error (MSE) of 1.96. However, the interpretability of the DNN results could be improved.

Rampášek et al. (2019) proposed the drug response variational autoencoder (Dr.VAE), utilizing a cooperative probabilistic approach with two outcomes. This method simultaneously models drug perturbation effects and drug response viability. It constructs a drug response predictor and a generative model of drug perturbation effects. The model also creates a functional low-dimensional representation in the latent gene expression space, enhancing the training data’s evidence. It achieved an AUROC of 0.71, a Pearson correlation of 0.89, and a P-value of 0.475. However, this model exhibits high computational complexity when the number of samples increases.

Su et al. (2019) showed a deep cascaded forest predictor called Deep-Resp-Forest that could figure out how drugs would work in both the GDSC and CCLE datasets. This model applies nonlinear map functions to transform raw features into vectors in a high-dimensional space. Subsequently, a deep cascade forest model learns features and predicts drug responses. The model achieves accuracies ranging from 93% to 98% and reduces time consumption by 300 s. However, this model does not provide exact sensitivity values since it poses an additional regression problem.

Sharifi-Noghabi et al. (2019) developed a DRP model called multi-omics late integration (MOLI) using DNN. This model integrates treatment data from various sources, including gene expression, aberration, and DRP. MOLI uses type-specific neural networks to combine features from each omics type into a single representation. It then uses DNN to learn this representation in order to predict how drugs will work. The model achieves an AUC of 0.8, surpassing existing models. However, it faces challenges related to class imbalance, data heterogeneity, and limited learning of combination data.

Liu et al. (2019) propose a deep convolutional network (DCN) for phenotypic DRP modeling. This computational model employs two convolutional neural networks in a pipeline. The first network extracts drug features from phenotypic screening data, while the second network extracts features for cancer cell lines. A fully connected network predicts the interaction between the drugs and the cancer cell lines. This model achieves mean coefficient of determination (R2) values of 0.826 and Pearson correlation coefficients of 0.909. However, it considers fewer features for cancer cell lines in small datasets.

Kuenzi et al. (2020) developed DrugCell, a DRP model utilizing a visible neural network (VNN). This model integrates different cellular systems as states of a mathematical model, which captures the drug’s structure and predicts the response. It incorporates human cells’ hierarchical structure, allowing drug patterns to be modeled in multiple cells. The method achieves a Spearman correlation of 0.8 and a high AUC of 0.83. However, this model does not consider specific critical mutations due to constraints in modeling the network structure.

Snow et al. (2020) introduced a DNN-based DRP model focused on androgen receptor (AR) responses, specifically for prostate cancer. This model learns the response of AR mutants and estimates the impact of androgen receptor drugs. It avoids the drug docking phase in feature construction, saving time and enhancing generalizability. The results demonstrate 80% precision, 79% recall, and 79% F1-score, with MCC values of 0.654, indicating satisfactory performance. However, this model primarily focuses on AR mutants and drugs similar to existing anti-androgens.

Wang et al. (2020) talked about a deep learning (DL) model for predicting drug metabolites that was made to fix the problems that regular machine learning algorithms have, like giving a lot of false positives and not being very accurate. This DL classification model extracts molecular fingerprints to make predictions. With a significant decrease in time complexity, the reported prediction accuracy is 78%. However, while this model can rank metabolites, it cannot predict the prevalence of metabolic sites. Furthermore, it still faces significant false-positive challenges.

Liu et al. (2020) developed DeepCDR, which utilizes a hybrid graph convolutional network for DRP. This network includes a uniform graph convolutional network and multiple sub-networks. It automatically learns the latent representation of topological patterns in cells and drugs. The model achieves a low RMSE of 1.058, a strong Pearson correlation of 0.923, and a Spearman correlation of 0.903. However, this model consumes more memory due to graph network formation.

Li et al. (2020) propose a deep learning-based drug response model that incorporates transcriptomic data and chemical structure information for drugs. A large dataset from the library of integrated network-based cellular signatures (LINCS) is used to train a DNN classifier. The model achieves high prediction accuracy, with an AUC of 0.89, when evaluated on non-small cell lung cancer data (NSCLC) from GDSC. However, extensive in vitro or in vivo investigations are required to validate the prediction results.

Zhang et al. (2021) utilized a DL model called consDeepSignaling, which is regulated by signalling pathways, to construct a DRP model. This model used gene expression and copy number variation data to investigate 46 signalling pathways. It extracted meaningful features from the signalling cascades to predict the sensitivity of cancer cell lines to drugs. When evaluated on multi-omics cancer data, the model achieved an MSE of 0.008 and a Pearson correlation of 0.85 for the testing data. However, the model is prone to overfitting as a significant performance difference exists between the training and test sets.

Kim et al. (2021) also developed a DRP model called DrugGCN using graph convolutional networks (GCN). This model constructed an undirected graph from a protein-protein interaction (PPI) network and mapped gene expression values to each gene as graph signals. The model implemented feature selection and used GCN to detect local features through localized filtering for accurate DRP predictions based on prior knowledge. The model achieved improved prediction accuracy, with a low RMSE of 2.5 and high Pearson and Spearman correlation values of 0.45 each. However, when applied to more significant input data, the model exhibits higher complexities as the plotting process requires more space and time.

Emdadi & Eslahchi (2021) developed the Auto-HMM-LMF model for feature selection-based DRP. This model employs a sparse autoencoder and a hidden Markov model with logistic matrix factorization. The autoencoder selects important features and processes copy number alteration data to extract crucial information. The model achieves 70% accuracy, 0.78 AUC, and 0.39 MCC coefficients for the GDSC dataset and 79% accuracy, 0.83 AUC, and 0.53 MCC coefficients for the CCLE dataset. However, the model exhibits high randomness in the prediction process.

Malik, Kalakoti & Sundar (2021) suggest a DNN framework that uses neighborhood component analysis (NCA) to help choose features from a mix of multiple omics data in DRP. This method utilizes the GDSC dataset for evaluation and achieves 94% accuracy for survival prediction. The model shows higher efficiency with an MSE of 1.154 and an overall regression value of 0.92. However, it introduces additional complexity for clustering to obtain binary responses and for hyperparameter optimization.

Li et al. (2021) developed the DeepDSC model using a deep learning approach for DRP. This model employs a stacked deep autoencoder to extract genomic features from gene expression data of cell lines and combine them with chemical features of compounds to generate the final response data. It achieves an R2 of 0.78, an RMSE of 0.23 for the CCLE dataset, and an RMSE of 0.52 and an R2 of 0.78 for the GDSC dataset. However, this model is limited as it is trained on a merged dataset involving GDSC, CCLE, and other inconsistent data.

Ma et al. (2021) present a deep learning-based DRP model to learn various features. It utilizes a flexible neural network architecture in cell lines that can be adapted to new contexts with minimal additional samples. The model adapts to changes in tissue types and transitions from cell-line models to clinical situations. It provides a Pearson correlation of 0.54 and an accuracy of 81%. However, this model only considers some vital features from the GDSC 1000 dataset.

Zhang et al. (2021) developed the AuDNN model, a synergistic drug combination-based DRP model assisted by a deep neural network. The model employs three autoencoders for different modeling tasks: gene expression, genetic mutation data, and copy number. The physicochemical properties of the drug are then put together with the outputs from the autoencoders and fed into the DNN model to guess how the drug combination will work on certain cancer cell lines. The model achieves 93% accuracy, 72% precision, 0.91 AUC, and 0.51 Kappa coefficients, with a minimized RMSE of 15.46 and a Pearson correlation of 0.74. However, this model introduces higher complexity in terms of processing cost. The GDSC and CCLE datasets and well-known drug data, such as IC50, are widely utilized for predicting anticancer drug sensitivity.

Nguyen et al. (2022) created GCN for DRP. They used drugs as molecular graphs to show how atoms are connected and cell lines as binary vectors of genomic errors. The model utilized convolutional layers to learn representative features from these graphs, which were then concatenated to generate drug-cell features. These features were used in a fully connected neural network for DRP. The model achieved low RMSE values of 0.0362 and a Pearson correlation of 0.8402 for the GDSC dataset. However, the model’s complexity could be improved.

She et al. (2022) presented a new project. The main objective of this project is to develop a deep learning approach called deep learning-based multi-drug synergy (DeepMDS), which utilizes various genomic data to identify novel synergistic drug combinations for a specific cell line. In order to formulate this methodology, we initially compiled a dataset consisting of gene expression profiles of cancer cell lines, target information for anti-cancer medications, and the reactions of these drugs against a diverse range of cancer cell lines. The model under consideration was constructed with the provided dataset and was founded on the concept of a fully connected feed-forward deep neural network. The regression task was highly successful, with an MSE of 2.50 and an RMSE of 1.58. The classification task was quite successful, achieving an AUC of 0.97, a sensitivity of 0.95, and a specificity of 0.93.

Tahmouresi et al. (2022) introduced a method for feature selection (FS) that combines gene rank with an enhanced Binary Gravitational Search algorithm (iBGSA). This combination is referred to as a pyramid Gravitational Search algorithm (PGSA). The authors noted that the proposed feature selection (FS) method demonstrated superior performance compared to previous wrapping strategies, resulting in a reduction of over 70% of the original number of features.

Shaban (2023) developed a hybrid feature selection technique (NHFSM) that uses filter and wrapper approaches to classify breast cancer patients. The technique uses information gain (IG), the hybrid bat algorithm, and particle swarm optimization (HBAPSO) to select nonrepetitive genes. The method achieved an accuracy of 0.97.

Alweshah et al. (2023) presented a new study. This study aims to simplify and computationally effectively solve gene selection problems in microarray gene expression data using the Black Widow Optimization algorithm (BWO) and the Iterated Greedy algorithm (BWO-IG). The hybridized BWO-IG technique improves local search capabilities, promoting more efficient gene selection. Tests on nine benchmark datasets showed that the BWO-IG technique outperforms the traditional BWO algorithm.

Yan et al. (2023) proposed a novel approach to overcoming the resistance of metastatic melanoma to immune therapies by targeting the upregulated chaperone-mediated autophagy (CMA) pathway in cancer cells. A prion-like chemical inducer named SAP was created to break down PD-L1 in a way that depends on chaperone-mediated autophagy (CMA). This restored the immune response against the tumor. This presents an immune reactivation strategy with clinical translational potential and serves as a reproducible example of precision medicine-guided drug development. SAP is a promising precision medicine method for treating melanoma that doesn’t respond to anti-PD1 therapy by using certain cellular mechanisms that are broken. However, changes in PD-L1 expression during treatment with anti-PD1 agents such as nivolumab may significantly impact patient outcomes.

Zhao et al. (2023) They created a glucose sensor based on a cell membrane (CMGS) to track GLUT1 transmembrane transport in tumor cells and look for GLUT1 inhibitors in traditional Chinese medicines (TCMs). The CMGS demonstrated high selectivity and stability in the presence of interfering molecules. The study also confirmed that the CMGS could couple with different cell membranes, showing higher current responses in tumor cell groups. The CMGS worked well with GLUT1-down-regulated cell membranes, showing that it could be used to track how glucose moves through different cell membranes. However, this technology still lacks comprehensive kinetic monitoring of other membrane proteins in addition to the effects of receptors on cell membranes.

He et al. (2023) proposed a cross-cohort computational framework to trace the tumor tissue-of-origin (TOO) of 32 cancer types based on RNA sequencing (RNA-seq). The framework uses logistic regression models to pick 80 genes for each type of cancer. This makes a total set of 1,356 genes based on transcriptomic data from 9,911 tissue samples from the 32 types of cancer whose TOO is known from the Cancer Genome Atlas (TCGA). The cross-validation accuracy of the framework reaches 97.50% across all cancer types. The model was tested on the TCGA metastatic dataset and the International Cancer Genome Consortium (ICGC) dataset, achieving an accuracy of 91.09% and 82.67%, respectively.

Wang et al. (2023) presented an explainable multimodal neural network (XMR) for predicting drug responses. It has two sub-networks: a visible neural network for learning genomic features and a graph neural network for learning structural features of drugs. The model is integrated into a multimodal fusion layer to model drug responses for gene mutations and molecular structures. A pruning approach is applied for better interpretations. The model uses five pathway hierarchies from the Reactome Pathway Database to predict drug responses in triple-negative breast cancer. The model outperforms other interpretable deep learning models in predictive performance and offers biological insights into drug responses.

Taj & Stein (2023) introduced the Multimodal Drug Response Prediction Program (MMDRP), a Python-based algorithm that uses a multimodal neural network to predict drug effectiveness on cell lines. The algorithm improves current methods by combining data from multiple cell lines, addressing skews in data, and improving chemical compound representation. as well as discover predictive biomarkers that help better match drugs to patients and guide the development of new drugs. However, these models require complex interpretation methods, which are done by analyzing the model after training. Furthermore, the interpretation can be model-specific or model-nonspecific.

Park, Lee & Nam (2023) They came up with drug response prediction models using ML and DL for 24 different drugs and then used the gene expression and mutation profiles of cancer cell lines to compare how well the models worked. There was no significant difference in how well the DL and ML models predicted drug responses for 24 drugs (RMSE ranged from 0.284 to 3.563 for DL and from 0.274 to 2.697 for ML; R2 ranged from −7.405 to 0.331 for DL and from −8.113 to 0.470 for ML). Among the 24 individual drugs, the ridge model of panobinostat exhibited the best performance (R2 0.470 and RMSE 0.623). Thus, we selected the ridge model of panobinostat for further application of explainable artificial intelligence (XAI).

Sharma et al. (2023) proposed a novel approach based on deep learning to utilize three distinct forms of multi-omics data for the purpose of predicting the efficacy of anticancer treatments in individual patients. The proposed DeepInsight 3D technique utilizes organized data to transform data into visual representations. Consequently, this enables convolutional neural networks to function. These networks are effective at handling inputs with high dimensionality and are capable of accurately modeling intricate interactions between variables. Nevertheless, the datasets’ exceptionally high dimensionality and the inadequately large number of annotated samples persistently impede these promising advancements.

Yue et al. (2023) presented a Java-based deep neural network technique named JavaDL. This technology utilizes chemical characteristics to forecast the response of cancer cells to various medicines. The approach employs an innovative cost function, regularization term, and early halting technique to mitigate overfitting and enhance the accuracy of the model. The treatment responses of multiple aggressive breast cancer cell lines were predicted using JavaDL, yielding very accurate and robust predictions with a r2 value as high as 0.81.

Liu & Mei (2023) proposed a novel drug sensitivity prediction model (NDSP) using deep learning and similarity network fusion approaches. It builds sample similarity networks and uses an improved sparse principal component analysis (SPCA) method to find drug targets in each omics dataset. The model uses three omics datasets and 35 drugs from the Genomics of Drug Sensitivity in Cancer. It achieves accurate sensitivity prediction of targeted and non-specific cancer drugs, benefiting precision oncology.

Sahu & Dash (2023) developed a hybrid multifilter-ensemble machine-learning model using grey wolf optimizer (GWO), recurrent neural network, and LSTM classifiers for classifying microarray cancer datasets. The model outperformed the GWO-RNN and GWO-LSTM in precision, recall, accuracy, and F1_Score metrics. The performance of the MF-GWO-RNN outperforms with an accuracy of 97.11%, 95.92%, and 92.81%, while the MF-GWO-LSTM outperforms with an accuracy of 97.17%, 98.56%, and 96.38% with leukemia and lung from the SRBCT datasets, respectively. Despite the good performance of this model, it has not been generalized to large datasets for other types of cancer. The various DL-based DRP methods discussed in the literature are summarized in Table 1.

Table 1 Summary of DL-based DRP methods in the literature.

Study	Model	Algorithm	Strengths	Limitations	Datasets	Results	
Chang et al. (2018)	CDRscan	Cancer drug response profile scan a novel deep learning model	High prediction accuracy	Low R2 values were found in a few GDSC compounds.	GDSC	The R2 value of 0.84 and AUROC value of 0.98	
Zhang, Chen & Li (2021)	ConsDeepSignaling	Deep learning model constrained by signaling pathways	Extracts the meaningful features with less complexity	Over-fitting problem	GDSC	MSE of 0.008 and Pearson correlation of 0.85	
Liu et al. (2019)	tCNNS	Twin convolutional neural network for drugs in SMILES format (tCNNS)	Two convolutional networks to extract features for cancer cell lines and drugs	Small training data and fewer features	GDSC	0.826 R2R2 and 0.909 Pearson correlation	
Nguyen et al. (2022)	GraphDRP	Graph convolutional networks for drug response
prediction	Deep representation of vital features	High complexity	GDSC	RMSE of 0.0362 and Pearson correlation of 0.8402	
Su et al. (2019)	Deep-Resp-Forest	Deep cascaded forest model, Deep-Resp-Forest	Multi-grained transformation of raw features	Does not provide the exact sensitivity values	GDSC
CCLE	93% to 98% accuracy and reduced time consumption of 300 s	
Zhang et al. (2018)	HNMDRP	Heterogeneous network-based method for drug response prediction	2% to 25% improvement of AUC	Poor incorporation of the cell line, drug, and target similarity network	GDSC	AUC-0.69 to 0.86	
Preuer et al. (2018)	DeepSynergy	Deep learning for drug synergies model	Maximal efficacy for combined representation of cell lines and drug synergies	Difficulties in generalizing the network when smaller drugs and cell lines	GDSC	Pearson correlation coefficient -0.73 and AUC-0.90	
Chen et al. (2018)	DBN and ontology fingerprints	Deep belief network and ontology fingerprints	High performance even when the data is unbalanced	Limited training capability	GDSC	The precision of 100%, 85% recall, and f-measure of 92%	
Matlock et al. (2018)	RF.	Random forest	Automatically lower the inherent bias	Stacking only with linear bias but does not consider nonlinear bias	GDSC	AUC of 0.9, error of 0.4, Eigen values as 0.95 and 0.23	
Xia et al. (2018)	ReNN	Recurrent neural network	Increased the response variance to 94%	Required hyper-parameter optimization for better tuning	GDSC	Pearson correlation of 0.972, Spearman’s rank correlation 0.965, R2R2 of 0.94.	
Tan et al. (2019)	Ensemble learning	Novel ensemble learning
method	Integrated the gene expression data signatures to improve prediction	It does not consider the cancer relationships from the sub-networks	GDSC	MSE 2.03	
CCLE	MSE 4.496	
Chiu et al. (2019)	DNN	Deep neural network	High accuracy due to pre-training with a large pan-cancer dataset	Limited interpretability.	GDSC	MSE 1.96	
Rampášek et al. (2019)	Dr.VAE	Drug response variational autoencoder	Improves the evidence of the training data	High complexity	GDSC	AUROC 0.71, Pearson correlation 0.89 and P-value 0.475	
Sharifi-Noghabi et al. (2019)	MOLI-DNN	Multi-omics late integration with deep neural networks	Optimize the representation of features for each omics type.	Class imbalance problem, data heterogeneity and limited learning of combination data	GDSC	0.8 AUC	
Kuenzi et al. (2020)	DrugCell using VNN	Visible neural network	High interpretation of cells	Does not consider some vital mutations	GDSC	Spearman correlation of 0.8 and high AUC of 0.83	
Snow et al. (2020)	DNN	Deep neural network	Omit drug docking to save time and generalise the model.	It is limited to mutants of androgen receptors.	GDSC	80% precision, 79% recall and 79% F1-score with MCC values of 0.654	
Wang et al. (2020)	DL-based drug metabolite prediction	Deep learning	High accuracy and reduced the time complexity	High false-positive problem	GDSC	Accuracy of 78%	
Liu et al. (2020)	DeepCDR	Hybrid graph convolutional network	Automatically learns the latent representation	Higher memory usage for the graph network formation	GDSC	Pearson correlation of 0.923, RMSE of 1.058, and Spearman correlation equal to 0.903	
Li et al. (2020)	DNN	Deep neural network	Large perturbation sample sets were used for training.	Validation requires large in vitro or in vivo experiments	GDSC, NSCLC	AUC of 0.89	
Kim et al. (2021)	DrugGCN	Graph convolutional network	High accuracy feature learning using past knowledge	High complexity for the larger graph plotting	GDSC	RMSE of 2.5, Pearson correlation of 0.45, and Spearman correlation values of 0.45	
Emdadi & Eslahchi (2021)	Auto-HMM-LMF	Autoencoder and hidden markov model	High accuracy	High randomness in the prediction process	GDSC	70% accuracy, 0.78 AUC and 0.39 MCC coefficients	
CCLE	79% accuracy, 0.83 AUC and 0.53 MCC coefficients	
Malik, Kalakoti & Sundar (2021)	DL with NCA	Deep learning with neighborhood component analysis	High accuracy in both DRP and survival prediction	Additional complexity for clustering to achieve binary responses	GDSC	Survival prediction accuracy of 94%, regression value 0.92, and MSE of 1.154,	
Li et al. (2021)	DeepDSC	Deep neural network for drug sensitivity in cancer	Less complexity	Limitation due to training on a merged dataset	GDSC	RMSE of 0.52 and R2R2 of 0.78	
CCLE	RMSE of 0.23 and R2R2 of 0.78	
Ma et al. (2021)	Few-shot learning	The few-shot learning framework bridges the many samples surveyed screens (n-of-many) to the distinctive contexts of individual patients (n-of-one)	High versatility	It does not consider all vital features	GDSC	Pearson correlation of 0.54 and accuracy of 81%	
Zhang et al. (2021)	AuDNNsynergy	Synergistic drug combination prediction by integrating multi-omics data in deep learning models.	Accuracy in predicting drugs combination responses to specific cancer cell lines	Higher complexity in terms of processing cost	GDSC	93% accuracy, 72% precision, 0.91 AUC, 0.51 Kappa coefficients, RMSE of 15.46, and Pearson correlation of 0.74	
She et al. (2022)	DeepMDS	A deep learning-based approach that integrated multi-omics data to predict novel synergistic multi-drug combinations	High performance with (RMSE) in the regression task, also gave the best classification accuracy, sensitivity, and a specificity	High complexity, Over-fitting problem	GDSC	(MSE) of 2.50 and (RMSE) of 1.58, the accuracy of 0.94, the sensitivity of 0.95, and a specificity of 0.93	
Tahmouresi et al. (2022)	PGSA	Pyramid gravitational search algorithm (PGSA)	A hybrid method in which the number of genes is cyclically reduced is proposed to conquer the curse of dimensionality, The PGSA ranked first in terms of accuracy with 73 genes	Classification of high-dimensional microarray gene expression data is a major problem	GEO
From NCBI	High accuracy (84.5%)	
Shaban (2023)	NHFSM	New hybrid feature selection method, hybrid method	New hybrid feature selection method, a hybrid method that combines the advantages of bat algorithm and particle swarm optimization based on filter method to eliminate many drawbacks	Validation requires large in vitro or in vivo experiments	GDSC	0.97, 0.76, 0.75, and 0.716 in terms of accuracy, precision, sensitivity/recall, and F-measure.	
Alweshah et al. (2023)	BWO-IG	Using two unique methodologies: the unaltered BWO variation and the hybridized BWO variant combined with the Iterated Greedy algorithm (BWO-IG)	The hybridized BWO-IG method is the best at doing local searches quickly and accurately.	These datasets contain a plethora of diverse and high-dimensional samples and genes. There is a significant discrepancy in the number of samples and genes present	GDSC	The key findings were an average classification accuracy of 94.426, average fitness values of 0.061, and an average number of selected genes of 2933.767.	
Zhao et al. (2023)	CMGS	Glucose sensor based on a cell membrane (CMGS) to track GLUT1 transmembrane transport in tumor cells and look for GLUT1 inhibitors in traditional Chinese medicines (TCMs).	The CMGS demonstrated high selectivity and stability in the presence of interfering molecules.	This technology still lacks comprehensive kinetic monitoring of other membrane proteins in addition to the effects of receptors on cell membranes.	TCMs	high selectivity and stability	
He et al. (2023)	TOO	Cross-cohort computational framework to trace the tumor Tissue-of-Origin (TOO)	A cross-cohort computational framework uses RNA sequencing to trace tumor tissue-of-origin (TOO) of 32 cancer types, utilizing logistic regression models for high accuracy.	Complexity, and limited learning of combination data	TCGA
ICGC		
Sahu & Dash (2023)	GWO-RNN and GWO-LSTM	Hybrid multifilter-ensemble machine-learning model using Grey Wolf Optimizer, Recurrent Neural Network, and LSTM	The performance of the MF-GWO-RNN outperforms with high accuracy with leukemia and lung from the SRBCT datasets	Difficulties in generalizing, and limited training	SRBCT	MF-GWO-RNN accuracy of 97.11%, 95.92%, and 92.81%, while MF-GWO-LSTM has an accuracy of 97.17%, 98.56%, and 96.38%, respectively.	

To determine the response to a drug, researchers analyze the primary tumor’s characteristics before treatment, including blood vessels, immune cells, and fibroblasts. Machine learning and deep learning frameworks can integrate these data to predict the response to specific treatments based on historical information. These predictive models are essential, especially when the test indicates that a tumor may not respond to existing therapies, prompting the exploration of alternative treatments. They provide valuable insights into predicting patient responses to treatment.

Understanding phenotypic drug response in cancer cell lines is crucial for discovering new anticancer drugs and repurposing existing ones. Researchers interested in phenotypic screening can access open data through the GDSC database to develop and evaluate their models. While earlier research focused on drug structures, the current analysis predominantly considers molecular fingerprints or the physicochemical properties of drugs.

Methodology

The review only looked at research publications published in English between 2017 and 2023. Figure 2 depicts a word cloud generated from the titles and abstracts of each article sampled using the VOS-Viewer 1.6.18 program. The visual representation in Fig. 2 conveys valuable information about articles focusing on DL models for cancer drug response. The circle sizes correspond to keyword frequencies, while the arcs indicate the strength of their correlations. Additionally, the color of the circles reflects the average annual frequency of these terms.

Figure 2 Keyword graph created from the titles and abstracts of the sampled articles.

Consequently, this illustration serves as an overview of the relevant articles in this area. We observed a strong association between specific keywords and the established grouping, including terms like machine learning and related concepts such as precision medicine, precision oncology, and personalized medicine. Moreover, the keyword pharmacogenomics significantly connects to deep learning within the graph. A distinct group, namely “deep learning and drug response and cancer and optimization,” was formed to examine research articles on optimizing deep learning models. This targeted search identified two relevant studies that incorporated the term optimization and its synonyms (Su et al., 2019; Liu, Shen & Pan, 2022; Pepe et al., 2022). This particular scenario further emphasizes the importance of this study, as it provides valuable insights and suggestions for utilizing optimized DL models in drug response prediction.

Cluster 1 (five items): Deep learning, Drug response prediction, Machine learning, Cancer, Precision medicine

Cluster 2 (three items): Precision oncology, Pharmacogenomic, Gene expression

Cluster 3 (two items): Personalized medicine, Drug response

This figure presents graphical visualizations of the co-occurrence of authors’ keywords in English research articles published between 2017 and 2023 in the journals. The chosen grouping was (Deep Learning AND Drug Response AND Cancer AND Optimization). Next, we searched PubMed scientific databases for studies with these terms in titles, abstracts, or keyword lists. The size of a node depends on its weight, and the link strength of a node determines its connective edges with other nodes. The sizes of the circles represent the frequency of keywords, the arcs represent the strength of their correlations, and the cluster indicates the average annual frequency of the terms.

The systematic literature review included 257 studies on DL, its optimization methods, and applications in various domains, including journal articles, conference proceedings, and book chapters published.

A total of 97 studies were excluded due to non-DL-relatedness, language barriers, or duplicate publications. After quality assessment, 160 studies were added to the review. This review aims to answer research questions and identify areas for further research.

In addition, key terms were adopted from the titles and abstracts of each cited article by calculating the weight of tf*df, also known as (term frequency–document frequency).

The tf*df weight, also known as term frequency—document frequency, is a quantitative measure that indicates the significance of a word within a document in a set of documents. It frequently serves as a weighting factor in the fields of information retrieval (IR) and text mining. The frequency of a word in the corpus counteracts the increase in the tf*df value in proportion to its occurrence in the document, so mitigating the influence of words that are generally more common. A basic ranking function can be calculated by adding up the word frequency multiplied by the document frequency for each query phrase. Many more advanced ranking functions are variations of this basic model.

The equation tf*df can be derived as:

tf∗df(t,d,D)=tf(t,d)×df(t,D)

where D is the overall number of documents in the corpus. The weights therefore tend to exclude popular terms, as they require a high term frequency within the given document and a low document frequency over the full collection of documents in order to have a high weight in tf*df. Given that the ratio within the logarithmic function of tf’s is consistently bigger than 1, it follows that the value of df (and tf*df) is greater than 0, as a term is present in a larger number of documents. The aforementioned notions were utilized to construct a database of keywords that can be employed to verify the frequency of keywords in each publication.

Hence, a total of studies were identified and thoroughly analyzed in order to address the following research inquiries: (Q1) to gain a comprehensive understanding of the concept of DL; (Q2) to uncover optimization strategies used in DL; and (Q3) to highlight the application areas and subjects related to DL. This study offers a contemporary direction for future research on the subject of deep learning for drug response prediction and precision medicine, facilitating more efficient progress in approaches and processes. The study conducted in this article demonstrates that research based on deep learning is experiencing significant growth, with a promising potential for development, implementation, and application in the future.

Research issue

Some recent studies show that optimizing parameters improves model performance considerably, demonstrating the need to optimize simple models rather than building complex models from scratch (Gao et al., 2020; Yousaf et al., 2022). If the problem is interpreted as an optimization problem, where the best configuration that maximizes the network performance is sought, a combination of search algorithms already available in the literature with a parameterized DL model can help to solve these problems. The specific problems are listed below: 1) Considering the use of DL tools for DRP, the user-defined hyper-parameters are often manually tuned or pre-fixed to a specified value to train the network (Wu et al., 2020).

2) Most DRP prediction models use biological information they already have, like pathway data, to get rid of features that are not important and make the models better. However, these drug response datasets are complex and have many dimensions (Zampieri et al., 2019).

3) The intricate nature of (DRP) problems necessitates the utilization of progressively intricate (DL) architectures, leading to a substantial augmentation in the number of learnable parameters, as observed by Baptista, Ferreira & Rocha (2023).

4) Learning continuous representations of compounds of the drug is limited since most methods need to pay more attention to the Graph convolutions (Sun, 2020).

Research questions

Deep learning models are highly complex and require much data to train, and artificial intelligence systems based on deep learning models have various deep structures with many hyperparameters.

However, there still needs to be more research on hyperparameters and structure settings for DL-based systems. Based on the previous research problem, the following research questions can be summarized:

Q1: What are the fundamental concepts related to deep learning algorithms that contribute to the development of precision medicine?

Q2: Which methodologies have been extensively utilized for the optimization of deep learning algorithms for drug response prediction?

Q3: What are the most common applications and domain areas where deep learning algorithms have been implemented?

Research objectives

The main goal of this work is to study the optimized DL-based models to predict the response of drugs to cancer treatments with superior performance to the methods recently developed in the literature.

Objective 1: This will help in understanding the primary knowledge regarding deep learning algorithms and what the real contribution is to the development of precision medicine.

Objective 2: It aims to identify optimization techniques involved in the training and learning of deep learning algorithms.

Objective 3: Investigate the studies that highlight the applications in the domain of deep learning algorithms.

Deep learning-based methods for drug response prediction

Deep learning is a subset of machine learning (ML) that uses artificial neural networks to imitate the learning process of the human brain. The ‘black box’ nature of ML and DL makes their inner workings difficult to understand and interpret. Deploying explainable artificial intelligence (XAI) can help explain why and how the output of ML and DL models is generated. As a result, understanding a model’s functioning, behavior, and outputs can be gained, reducing bias and error and improving confidence in decision-making (Love et al., 2023).

Although the theoretical exploration of DL dates back to the 1960s, limitations imposed by the limited computational capabilities in use at the time may have improved the advancement. DL studies gained new impetus due to advances in processor units, enabling information processing in the era of big data. The first DL architectures were proposed in 2006, and since then, research on DL methods has gained significant momentum. DL models have been harnessed to develop learning health systems that continuously learn from clinical patient databases and update themselves as new data becomes available (Boldrini et al., 2019). Future research might address some of the current shortcomings in the use of DL. For instance, DL models may overfit certain datasets and fail to generalize to new data. More research is required to develop DL models that successfully generalize to various situations and data. Since deep learning models can be difficult to read and comprehend, their practical applications are rather limited. There is a need for more research on the interpretability and transparency of DL models, as well as the development of methods for expressing and elucidating the decision-making processes of these models (Kumar et al., 2023). DL has found extensive applications in constructing Artificial Intelligence (AI) models across various domains. These DL-based techniques have proven instrumental in solving complex problems by leveraging computational resources to extract valuable insights from vast databases, enabling the creation of more compact and intelligent models. Figure 3 illustrates the components of the DL family (Alzubaidi et al., 2021).

Figure 3 Deep learning family.

According to the findings of Salleh, Talpur & Talpur (2018), machine learning models are initially trained using raw data during the classification phase. Subsequently, the learned information is utilized to classify new observations. Traditional classification models like naive Bayes (NB), support vector machine (SVM), and K-nearest neighbor (KNN) have been extensively employed in the literature to address classification challenges. However, it is essential to note that these models were not explicitly designed to handle data inaccuracies or uncertainties in processing. This study, on the other hand, shows new deep neural fuzzy systems models that combine adaptive neuro-fuzzy inference systems (ANFIS), deep neuro-fuzzy systems (DNFS), and deep neural network (DNN) classifiers. The study outlines the general structures of these models and elucidates the underlying principles guiding these classifiers.

Adaptive neuro-fuzzy inference systems (ANFIS): In real-world classification problems, it is evident that nonlinear behavior is prevalent. Therefore, the modeling and representation approaches to address these problems must incorporate these nonlinear characteristics. To improve the classification performance in highly nonlinear scenarios, researchers have utilized adaptive neuro-fuzzy inference systems (ANFIS) (Karaboga & Kaya, 2019). 1) Deep neural network (DNN): A DNN is constructed as a sequential arrangement of multiple layers comprising neurons that serve as nonlinear units for processing information. The inherent nonlinearity arises from the interaction between activation potentials in different layers. This leads to the representation of nonlinear functions in subsequent layers, resulting in a complex mapping from input to output. The training process involves iteratively presenting the data to the network, which establishes this mapping. Moreover, the connections between neurons in the network are represented by weights, which are adjusted for each neuron using a learning technique. The most widely recognized method for weight adjustment is error backpropagation (Karaboga & Kaya, 2019). However, there are various algorithms for training networks.

2) Deep neuro-fuzzy systems (DNFS): Deep learning (DL), especially through DNN, has become a popular way to model difficult problems that are not symmetrical and have noise from outside sources. This recent advancement has presented a dominant approach to addressing such challenging scenarios. As Talpur et al. (2023) say, one important thing that makes DL models work is their ability to learn from different datasets, which lets them learn different levels of complexity in large-scale tasks.

The primary goal of this study is to propose the utilization of optimized artificial intelligence (AI) approaches in modern medical science. The study also aims to explore the potential benefits of employing optimization strategies, which can serve as a practical and advanced step towards improving the accuracy of drug response prediction in the context of cancer. Table 2 presents an overview of different deep learning methods developed over the past decade and a comparative analysis. Some of the strategies in this table are employed in big data analytics and healthcare applications. Furthermore, the application summary displays a wide selection of general solutions.

Table 2 Comparison of the most popular deep learning methods.

Deep learning
algorithms	Description	Strengths	Weaknesses	
Denoising autoencoders
Chi & Deng (2020)	Designed to handle corrupted data.	High ability to extract useful features and compress information.	High computational cost and addition of random noise. Scalability issues in problems with high dimensions.	
Sparse autoencoder
Munir et al. (2019)	Model for data reconstruction in a high sparsity network where only part of the connections between neurons is active.	Linearly separable variables are produced in the encoding layer with ease.	High computational training time is required for processing input data.	
Restricted Boltzmann
machine (RBM)
Sarker (2021)	A stochastic recurrent generative neural network is one of the first capable of learning internal representations.	The ability to create patterns with missing data and to solve complex. combinatorial problems	The training process is complex and time-consuming due to the connectivity between neurons.	
Deep Boltzmann
machine (DBM)
Wang et al. (2019)	Boltzmann network with connectivity constraints between layers to facilitate inference and learning	Allows robust extraction of information with the possibility of supervised training enabling the use of a feedback mechanism	High training time for large datasets and difficult adjustment of internal parameters	
Deep belief network
(DBN) Wang et al. (2018)	Designed with undirected connection in the first two layers and layers and direct link in the remaining layers	Ability to extract global insights from data, performing well on dimensionality reduction problems	Slow training process due to the number of connections	
Convolutional neural network (CNN) Liu et al. (2019)	A deep neural network structure inspired by the mechanisms of the biological visual cortex.	Allows different variations of training strategies with good performance for multidimensional data and ability to the abstract representation of raw data	A large volume of data with more hyperparameter tuning is required to perform well.	
Recurrent neural
Network
Mirjebreili, Shalbaf & Shalbaf (2023)	Framework designed for modelling sequential time series data through a time layer to learn about complex variations in the data	Most widely used for modelling time-series data.	The training process is complex and sometimes affected by vanishing gradients. Many parameters must be updated, making the learning process time-consuming and difficult.	

Convolutional neural network

Due to their impressive performance, CNNs, or convolutional neural networks, have gained widespread usage in various fields (Kumar et al., 2020). These networks differ from traditional neural networks’ approaches to processing and extracting information. The convolution operation, a critical mathematical process in CNNs, enables complex data analysis in convolution operations at different layers. The networks effectively extract and represent information in diverse patterns. A typical CNN structure is illustrated in Fig. 4. The network consists of concatenated layers, including the convolution, pooling, fully connected, and output layer responsible for sample classification (Alzubaidi et al., 2021).

Figure 4 The structure of the convolutional neural network.

CNNs are the most prevalent type of deep neural network, named after the convolution operation, which involves linear mathematical operations performed on matrices. A CNN incorporates multiple levels, such as convolutional, non-linear, pooling, and fully connected layers. These layers possess adjustable parameters that require configuration. CNNs exhibit exceptional.

Performance in various machine learning tasks, particularly in applications involving image data, such as image recognition datasets, computer vision, and natural language processing (Alyasseri et al., 2022). CNN networks were initially developed for image processing tasks and have shown remarkable performance in tasks like hand gesture classification (Gadekallu et al., 2021). Conventional networks face challenges when dealing with high-dimensional data, such as images, due to many parameters and issues like overfitting (Spadea et al., 2019). However, several studies have applied CNN networks to other data types despite being initially designed for image data. These include diverse data such as electrical signals, time series datasets, and others (Currie et al., 2019; Chriskos et al., 2021; Cole et al., 2021).

A CNN is a type of feedforward neural network comprising four layers. The first layer is the input layer, which receives the samples from the dataset. The second layer is the convolutional layer, which plays a crucial role in CNNs by performing mathematical convolution operations (Xue, Dou & Yang, 2020). Unlike conventional neural networks that rely on matrix multiplication, CNNs utilize convolutional operations. The output of the convolutional layer is then connected to a global maximum pooling layer. Pooling layers are commonly employed after convolutional layers to simplify the information within the output. Typically, pooling units identify the maximum activation within their respective regions. Finally, the fully connected layer processes the pooled output to produce the result.

Deep learning neural networks

Deep learning techniques have significantly influenced neural network applications in recent years. Deep learning structure involves adding multiple hidden layers to conventional neural networks. While adding hidden layers is straightforward, the operations performed within these layers introduce increasing complexity to the models. Hinton & Osindero (2006) initially introduced the concept of deep learning through a greedy multilayer learning algorithm. This algorithm involves pre-training the neural network using an unsupervised learning technique before proceeding with the conventional layer-by-layer training process (Sze et al., 2017). Deep learning methods are renowned for handling large datasets and utilizing mathematical algorithms that minimize the need for random numbers in the initial stages of unsupervised learning. Consequently, the network exhibits improved performance after training under these conditions. Various types of deep learning tools are available, and Fig. 5 provides a general illustration of the structure of a deep neural network comprising input, output, and multiple hidden layers.

Figure 5 The structure of the deep learning neural network.

The utilization of a multilayer deep learning approach is pervasive across diverse applications. This methodology involves the training of hidden layers through the application of the backpropagation algorithm and gradient descent. The network architecture is organized with an input, an output, and multiple hidden layers, each comprising numerous neurons functioning as information processing units. By manipulating the connections between the input and hidden layers, the algorithm generates novel variables for each neuron, which are subsequently transmitted to the output layer. The output layer then predicts outcomes based on the provided input data (Fu, Lv & Zhang, 2022). The efficacy of the backpropagation algorithm is contingent upon the network architecture and the types of incorporated layers (Kuninti & Rooban, 2021).

Regarding neural network architecture, the complexity of the chain rule can vary, even though the derivative operations remain the same. This variance in complexity can lead to gradient landscapes with varying smoothness. For instance, feedforward neural networks can be formulated as matrix and vector operations, making derivative calculations straightforward. Recurrent neural networks require additional steps for gradient computation during forward and backward passes. On the other hand, deep neural networks with multiple layers can introduce many network weights. Utilizing backpropagation algorithms in high-dimensional spaces can lead to performance degradation. Efforts have been made to address these challenges by devising improved approaches, such as Adam’s algorithm, which specifically targets the training of large neural networks. Specific layers, like the dropout layer, serve as explicit regularizers. During training, the dropout layer randomly sets some weights to zero, operating as a stochastic regularizer. Adding dropout can stop training algorithms (like backpropagation) from rushing to local minima of the loss function too soon, which has a direct effect on how well the training process works.

Other fast convergence training algorithms were made using mathematical models that were specifically designed for different situations, like when there are large databases or multimodal training surfaces. A key differentiating factor in deep learning lies in the highly non-linear relationships among the various layers of the network, enabling the handling of functions with varying degrees of complexity. Consequently, deep networks can recognize intricate patterns and effectively model complex problems (Zemouri et al., 2020).

Adaptive network-based fuzzy inference system

ANFIS models, which are another name for neuro-fuzzy systems, use computer programs that mix the uncertainty of fuzzy systems with the modeling flexibility of artificial neural networks (Farhadi et al., 2020). In ANFIS models, the adaptive topology of neural networks and fuzzy logic (Lima-Junior & Carpinetti, 2020) are put together to make a strong and complete method. The ANFIS model leverages the strengths of fuzzy logic and artificial neural network methods, addressing limitations that arise when these two approaches are used independently. One interesting thing about the ANFIS model is that it can find the best values for the rule base and membership functions so that it can accurately model systems that are full of unknowns. Within the ANFIS framework, the internal parameters that define the rule base and membership functions are optimized to hold the fuzzy logic that describes how input and output work together in a system. This optimization process is executed through the utilization of neural network training algorithms. Similar to how neural networks are trained, the main goal is to make the fuzzy system better at learning based on the available dataset (Farhadi et al., 2020).

This will allow the model to adapt to the system that is being studied. The ANFIS model comprises two distinct functions: the premise and the consequence. Each part relies on internal parameters that capture the inherent uncertainty within the model. These parameters are determined through network training, employing optimization algorithms to identify regions with minimal error. Throughout the training process, the input data undergoes iterative presentation to the adaptive neuro-fuzzy inference system (ANFIS) network. According to Roger Jang (1993), after the training phase, the acquisition of fuzzy IF-THEN rules occurs through the connection of the consequence and antecedent components. The computational architecture of the ANFIS model is characterized by five distinct layers. The initial layer encompasses four membership functions and four rules. Figure 6 shows a visual representation of this five-layered structure, illustrating how Claywell et al. (2020) implemented the Takagi-Sugeno model for the fuzzy system within the neuro-fuzzy network.

Figure 6 Architecture of the adaptive network-based fuzzy inference system (ANFIS).

A series of distinct layers characterize the architecture of the ANFIS. The initial layer, denoted as the fuzzification layer, organizes input data into fuzzy clusters. Subsequently, the second layer, the ruler layer, generates firing strengths based on the values computed in the fuzzification layer. The third layer, termed the normalization layer, undertakes the normalization of the firing strengths associated with each rule, employing diverse methodologies for this purpose. The fourth layer, defuzzification, executes defuzzification operations, thereby converting fuzzy rules into tangible numerical values. The final step is the fifth layer, which is called the summation layer and is the output layer. It takes rule outputs from the defuzzification layer and adds them up to get the ANFIS output. The application domains wherein ANFIS hybrids find utility in addressing real-world challenges are visually depicted in Fig. 7.

Figure 7 Some application areas use ANFIS hybrids.

Electronics, Mechanical, Geological, Meteorology, Biomedical, Civil Engineering, Physics, Computer Science, Business, Agriculture.

Using deep learning workflows to predict drug response

Over time, workflows have become integral to developing bioinformatics, computational biology, and computer-aided medicine solutions. They aim to simplify and automate repetitive decision-making tasks, minimizing errors and increasing efficiency. Workflows are crucial in facilitating faster and more informed decisions while enabling researchers to collaborate proactively and with greater agility. Although deep learning workflows have gained prominence in recent years, they have advantages and disadvantages when utilized to create rapid and innovative solutions. The main advantages of employing deep learning workflows can be summarized as follows: Deep learning workflows offer the advantage of automatically labeling datasets or generating reports, thus facilitating effective communication among researchers.

These workflows improve prediction accuracy by automatically retraining models with new data as it becomes available, ensuring that predictions remain accurate as the data evolves.

Deep learning workflows are adept at saving time by automatically parallelizing the training process across multiple graphical processing units (GPUs), thereby accelerating model training.

However, there are potential disadvantages associated with workflows, as highlighted in the existing literature:

Inaccurate formatting or data organization can pose challenges during the training process.

Workflows may have limited flexibility in assessing new scenarios incorporating models and requirements demanded by medical professionals or in weighing specific treatment characteristics.

Deep learning models built on workflows can be difficult to interpret, making comprehending the rationale behind specific predictions challenging. From a technical point of view, workflows require maintenance by constantly updating their flows to take full advantage of advancing hardware and software resources.

Despite the disadvantages mentioned above, utilizing deep learning workflows offers significant benefits by enabling collaboration between medical specialists and developers. It is very helpful to combine these two areas, especially when it comes to solving drug response prediction (DRP) issues that need quick and accurate answers, since creating new cancer treatments costs a lot of money. Most drug response prediction workflows follow a general set of steps illustrated in Fig. 8. These steps include:

1) Selecting an appropriate DL framework for implementing the model.

2) Defining the specific problem, such as predicting drug sensitivity or drug synergy.

3) Choosing the dataset(s) for model training (refer to Table 3 for available dataset resources).

4) Determining the types of input data to be utilized.

5) Managing multiple data types, wherein users must decide whether to concatenate all features regardless of their type or adopt a multimodal strategy that employs separate subnetworks for each type.

6) Choosing the right data representations and preprocessing methods are part of designing the model architecture. The DL architectures are made to fit the type of data that comes into each network.

7) Tuning hyperparameters and training the model, wherein hyperparameter optimization can be performed manually by exploring all or a subset of the possible combinations of user-specified values or utilizing other optimization techniques.

8) Evaluating the model output by selecting suitable scoring metrics, defining the process of splitting the screening dataset into training and validation/test sets, and identifying any external datasets to be used for model validation.

9) Providing explanations and interpretations of model predictions.

Figure 8 Cell lines, somatic mutations, copy number variations, and gene expression data are familiar input data sources for data-driven drug response prediction models.

Other omics data (epigenomics, proteomics, and so on) can also be included in the models. Gene expression traits are continuous traits that have been normalised, while somatic mutations are usually binary (change present or absent). When target information is used, it is frequently used to extract features that reflect the routes associated with a specific goal. Values that describe dose-response relationships are the outputs of drug response prediction models. For single drugs, it is usually half-maximal inhibitory concentration (IC50), 50% growth inhibition (GI50), or area under the dose-response curve (AUC). For drug combinations, the output variable is often a score that measures how the combination of drugs affects a given reference model. Half-maximal inhibitory concentration (IC50), 50 per cent growth inhibition (GI50), or area under the dose-response curve are commonly used for single drugs (AUC). The output variable for drug combinations is often a score that quantifies the drug combination’s effects using a reference model.

Table 3 Data set resources from pan-cancer screening studies Liu et al. (2020) and Malik, Kalakoti & Sundar (2021).

Data set	Link	
CCLE	https://portals.broadinstitute.org/ccle	
CTRPv1	https://portals.broadinstitute.org/ctrp.v1/	
CTRPv2	https://portals.broadinstitute.org/ctrp/	
EGA	https://www.ebi.ac.uk	
GDSC1	https://www.cancerrxgene.org/	
GDSC2	https://www.cancerrxgene.org/	
GEO	https://www.ncbi.nlm.nih.gov/geo/	
ICGC	https://dcc.icgc.org	
METABRIC	https://www.mercuriolab.umassmed.edu/metabric	
TCGA	https://portal.gdc.cancer.gov	

New trends and directions on dl for cancer drug response prediction

Hyperparameter optimization of deep learning methods

Optimization techniques have continuously evolved and found applications in various domains, including medication response prediction. These techniques are now used to select models for machine learning and deep learning tasks. Figure 9 provides a classification of optimization approaches. Population-based, nature-inspired algorithms, in particular, have made significant advancements over the past two decades and are extensively utilized for model selection and hyperparameter optimization. Evolutionary algorithms, among the population-based nature-inspired algorithms, have garnered attention from researchers due to their simplicity of implementation and superior problem-solving capabilities.

Figure 9 Overview of optimization algorithms.

Figure 10 presents a schematic framework for a DL hyperparameter optimization cycle. The cycle shown in the figure can be conducted by an evolutionary algorithm that guides the search based on a measure calculated on the dataset. As the algorithm searches to minimize this measure, increasingly better models are created.

Figure 10 Evolutionary-assisted deep learning schematic workflow.

Evolutionary computation is a paradigm that derives solutions from natural concepts rather than mathematical principles. It encompasses the implementation of an evolutionary algorithm, which aims to find potential solutions within a search space (Spirov & Myasnikova, 2022). The standard evolutionary algorithms can be described as follows:

Bayesian Optimization (BO): BO is employed for discovering optimal parameters that maximize or minimize an unknown objective function. It relies on a probabilistic surrogate model and an acquisition function. The surrogate model estimates the objective function, and the acquisition function guides the selection of the next evaluation point. By leveraging this approach, the optimal solution of the objective function can be determined with fewer evaluations and less redundant sampling (Young et al., 2020).

Particle Swarm Optimization (PSO): In PSO, a population of particles moves through the search space under the direction of their individual flight experiences. The fitness value function is defined to identify an ideal particle solution within the entire search space. PSO metaphorically represents the behavior of particles in finding optimal solutions (Freitas, Lopes & Morgado-Dias, 2020).

Differential Evolution (DE): DE is a stochastic, straightforward, and efficient evolutionary algorithm (Qaddoura, Faris & Aljarah, 2021). It combines population-based global search techniques with a simple mutation operation to minimize genetic complexity. DE’s flexibility allows for the dynamic tracking of the ongoing search, enabling adjustments to the search directions for global convergence and robustness (Ahmad et al., 2022). The ant colony optimization (ACO) metaheuristic approach Wang & Han (2021) introduced was introduced to address discrete optimization problems. This algorithm emulates the foraging behavior of ants in search of food, representing candidate solutions. Ants randomly explore their environment around the colony, examining and evaluating food sources (objective function). Upon returning to the colony, an ant deposits pheromones on the path based on the quality of the solution. The pheromone serves as a communication mechanism, indicating the quantity and quality of the food source and attracting other ants to explore the respective candidate solution for the problem.

The artificial bee colony (ABC) algorithm is a swarm-based metaheuristic optimization method (Kaya, 2022) inspired by the foraging behavior of honeybees. The algorithm consists of forager bees, observer bees, and scouts. Each food source is assigned a forager bee, which communicates the location of food sources to observer bees. Scouts search the area, and worker and spectator bees use the scouts’ information to exploit the food sources.

Grey wolf optimization (GWO) is a novel swarm intelligence-based optimization algorithm (Albadr et al., 2022) that mimics the predatory behavior of grey wolves. The algorithm imitates the cooperation mechanism among wolves to achieve the desired optimization results.

Yang & Deb (2009) developed Cuckoo Search (CS), which draws inspiration from the parasitic behavior of some cuckoo species and the Lévy flight pattern seen in bird foraging. Some cuckoos deposit their eggs in the nests of other birds, selecting nests with recent egg production and removing existing eggs to increase their chances of hatching. CS utilizes this behavior, along with Lévy flights, to explore the search space for optimal solutions (Joshi et al., 2017).

Cheng & Xiong (2023) developed a multi-strategy adaptive Cuckoo Search CS (MACS). MACS uses a parameter control strategy, three search strategies, and a probability matching scheme. Experiments on 42 benchmark problems show MACS outperforms other algorithms, demonstrating its versatility and robustness. The Cuckoo Search (CS) algorithm is a popular and efficient search technique for tackling numerical optimization problems.

The FA emulates the behavior of fireflies and uses point representations in the search space. It effectively handles multimodal functions by simulating the attraction and movement of fireflies during the search process. FA incorporates brightness and appealing brightness, where brightness determines the firefly’s location and appealing brightness influences the firefly’s distance, resulting in optimized objectives (Nayak et al., 2020).

The Firefly Algorithm (FA) The Firefly algorithm is one of the known metaheuristic algorithms used in a variety of applications. Metaheuristic algorithms are successful methods of optimization. Recently, a new and efficient version of this algorithm was introduced as NEFA, which indicated a good performance in solving optimization problems (Rezaei & Rezaei, 2022).

In the continuous development of nature-inspired optimization algorithms, evaluating newly developed algorithms and their performance in hyperparameter search is crucial. Recently developed bio-inspired optimization algorithms that can be used are the dingo optimization algorithm (DOA) (Peraza-Vázquez et al., 2022), golden eagle optimization (GEO) (Mohammadi-Balani et al., 2021), black widow optimization (BWO) (Hayyolalam & Pourhaji Kazem, 2020), wild horse optimization (WHO) (Naruei & Keynia, 2021), and the Harris hawk optimization (HHO) (Heidari et al., 2019). For detailed implementation in MATLAB, please refer to Table 4.

Table 4 Codes for Bio-inspired optimization techniques are freely available in MATLAB.

Bio-inspired optimization techniques	URL	References	
Dingo optimization	https://www.mathworks.com/matlabcentral/fileexchange/98124-dingo-optimization-algorithm-doa?s_tid=srchtitle	Peraza-Vázquez et al. (2021)	
Golden eagle optimizer	https://www.mathworks.com/matlabcentral/fileexchange/84430-golden-eagle-optimizer-toolbox?s_tid=srchtitle	Mohammadi-Balani et al. (2021)	
Black-widow optimization	https://www.mathworks.com/matlabcentral/fileexchange/94080-black-widow-optimization-algorithm	Hayyolalam & Pourhaji Kazem (2020)	
Wild horse optimizer	https://www.mathworks.com/matlabcentral/fileexchange/90787-wild-horse-optimizer	Naruei & Keynia (2022)	

Neural architecture search

Due to its strong learning capabilities, deep learning is highly effective in various domains. However, constructing neural architectures relies on researchers’ existing knowledge and expertise. This multitude of possibilities can make it challenging for beginners to make appropriate adjustments to the architecture based on specific requirements. Additionally, preconceived knowledge and rigid thinking patterns might hinder the emergence of new neural architectures (Miriyala et al., 2022). Neural Architecture Search (NAS) has been developed to discover unique and insightful architectures in this context.

NAS is a research field that aims to harness the capabilities of neural networks by finding the optimal architecture for specific needs, particularly in drug response prediction. Instead of relying on manual tweaking of neural networks by human experts, NAS automates this process, allowing for the discovery of more complex and accurate architectures. It has emerged as a promising area of study, offering several advantages, such as saving time in model development, reducing costs associated with high-end GPUs on public clouds, and enabling the exploration of multiple alternative models (Baymurzina, Golikov & Burtsev, 2022). Figure 11 provides an overview of the NAS search framework.

Figure 11 NAS strategy framework.

When building deep neural networks, it can be hard to set up hyperparameters that don’t have anything to do with the architecture. These include activation functions, dropout rates, convolution filters, and training algorithms. This is because these parameters have a big effect on how well the networks work. To overcome this, prominent NAS studies in recent years have aimed to automate the selection of these hyperparameters, as summarized in Table 5. Designing all possible combinations of non-architectural hyperparameters and architectural choices is laborious and time-consuming (Dai et al., 2020). Hence, it often leads to excessive computational resource usage spent on parameter adjustment instead of training the algorithm.

Table 5 Summary of neural architecture search paradigms for DL.

Study	Model	Strengths and limitations	
Baker et al. (2016)	MetaQNN	The neural architecture selection process is modelled as a Markov decision process.	
Liu et al. (2021)	Large-scale evolution	Evolutionary algorithms are used to learn the best neural architecture automatically.	
Liu et al. (2021)	GeNet	Agent-based neural architecture search	
Xie & Yuille (2017)	NAS-RL	Reinforcement learning-derived neural architectures	
Zoph & Le (2016)	NASNet	Using the concept of artificial neural architecture design proposes a modular search space	
Zoph et al. (2018)	GDAS-NSAS	Weight sharing is implemented to train a new neural architecture sequentially.	
Zhao et al. (2021)	NAS-RUL	A gradient-based neural architecture search method	
Alyasseri et al. (2022)	NAS-Bench-NLP	Search based on natural language processing (NLP)	
Shi et al. (2022)	Genetic-GNN	Evolutionary algorithms are used to learn the best neural architecture automatically.	
Ding et al. (2022)	NAP	Neural architecture search with pruning	
Mun, Ha & Lee (2022)	DE-DARTS	Neural architecture search with dynamic exploration	

Architectural parameters, however, directly affect the network’s topology, including the number of layers and neurons and the arrangement of connections between neurons. The topology of a neural network, or how its neurons are interconnected, plays a crucial role in its functioning and learning processes. Early methods attempted to simultaneously search for hyperparameters and architectures, but they frequently had limited search spaces and small datasets (Wu et al., 2019; Liu, Shen & Pan, 2022).

NAS methods automatically find optimal neural networks without human assistance. Also, the NAS can potentially eliminate the trial-and-error process of manually designing neural architectures for deep learning development (Mun, Ha & Lee, 2022).

Transfer learning

Transfer learning is a technique in deep learning that allows knowledge acquired from one task to be applied to a different but related task. In the drug response prediction (DRP) context, transfer learning is particularly beneficial. It leverages knowledge obtained from similar problems, which is readily available, to optimize computational resources during model training. This approach saves time in data collection and processing while also assisting specialists in making faster decisions by generating prompt responses.

The goal of transfer learning is to enhance the learning process of a target task by leveraging knowledge from a source task. Since it can be challenging to gather sufficient data-driven models, transfer learning offers the potential to utilize cell lines as a source for predicting target model outcomes. By utilizing a large dataset to train the target task, transfer learning addresses the limitations of small datasets, accelerates the learning process, and improves accuracy. Previous studies have demonstrated that transfer learning can be instrumental in reducing overfitting (Zhou, Pan & Tsang, 2019; Celik et al., 2020).

Transfer learning lets you use a network that has already been trained and has learned many low-level features from layers that are closer to the samples in the drug response dataset. Even though the datasets may not be identical, the low-level features acquired by the original deep neural network in the larger dataset are generally comparable to the low-level features in the smaller target dataset (Hosseinzadeh Kassani et al., 2022). On the other hand, high-level attributes in the final layers of the network focus on complex characteristics crucial for distinguishing classes. Transfer learning makes training massive deep networks more feasible, yields more promising results, and is significantly more cost-effective than training a deep neural model from scratch (Mirjebreili, Shalbaf & Shalbaf, 2023). When training a network from scratch, the weights are initialized arbitrarily, whereas transfer learning involves transferring the weights learned in a different domain to the medical domain. Figure 12 illustrates a typical transfer learning approach for drug response prediction (DRP).

Figure 12 Transfer learning approach for DRP.

Tansey et al. (2022) suggested a deep transfer learning model for predicting drug sensitivity. This model uses transfer learning algorithms to combine multi-omics data from a single cell. Their approach combines information from extensive bulk sequence data with the diverse landscapes of single-cell sequencing. Similarly, Zhu et al. (2020) developed an ensemble transfer learning framework that improves the performance of existing techniques by incorporating prediction patterns learned from related data. The main aim of transfer learning is to create a highly effective learner for a target domain with limited available data by utilizing prediction patterns learned from a similar source domain with abundant data. There have been recent efforts to utilize transfer learning for (DRP) (Zhu et al., 2020; Chen et al., 2022), which opens up opportunities for researchers to contribute to the development of novel transfer-based approaches (Ardalan & Subbian, 2022).

Discussion

The literature demonstrates that DL-based models for DRP generally outperform traditional ML models. These DL methods can be further explored to enhance prediction performance. Some DL models have achieved high correlations when predicting drug responses for unknown drug-cell line pairs. However, DL-based models still face challenges in generalizing to new cell lines and drugs. Moreover, there needs to be more usage of faster DL models like CNN or appropriate models like RNN or long short-term memory (LSTM). The application of NLP-inspired techniques in DL models is also relatively rare. In order to improve the NLP-based models, specific medical databases can be employed as dictionaries instead of standard ones. A successful DL model in DRP will likely depend on the effective combination of learned representations and carefully selected manual features. It is, therefore, crucial to harness the potential of DL-based DRP models and utilize them in impactful research studies.

Ensembles of classifiers have become increasingly common in DRP to reduce model uncertainty and improve generalization performance (Tan et al., 2019). Developing methodologies for generating potential ensemble members is essential to creating more robust classifiers. A well-constructed ensemble is one where individual classifiers within the ensemble predict different regions of the input space, enhancing the final model’s overall robustness. Moreover, combining identical classifiers only sometimes improves performance (Xiao et al., 2018; Zhu et al., 2020).

Deep learning networks can sometimes involve millions of adjustable coefficients, making searching for the best architecture and corresponding hyperparameters challenging. Efficient algorithms that explore the search space and intensify in promising regions should be implemented in research. Evolutionary algorithms enable obtaining a population of solutions, which can advance the model’s robustness by combining the population into ensembles. From a professional standpoint, there are numerous potential research directions and contributions when considering computing hardware developments for deep learning models in drug response prediction (Ploug & Holm, 2020; Tjoa & Guan, 2021; Salahuddin et al., 2022; Jin et al., 2022).

This article argues that deep learning models can enhance medical diagnosis and therapy planning as research progresses in various medical domains. However, deep learning models often function as black-box models, needing more interpretability, which makes decision-making challenging. It is hard to fix differences in diagnoses that show big differences between patients, biases, or mistakes in diagnosis if you don’t understand or explain AI diagnostics (Holzinger et al., 2019; Salahuddin et al., 2022; Jin et al., 2022).

Xue & Chuah (2019) suggested an explainable deep learning-based medical diagnostic system that explains the outcomes of the diagnosis. Researchers and developers have a critical responsibility to ensure that biases in AI decision-making can be minimized by evaluating multiple datasets before the clinical deployment of an AI system. While eliminating all biases may not be possible, the goal is to reduce them. As a result, self-modifying AI systems can be developed and integrated into routine healthcare settings without compromising the essential requirements of contestability.

Efficient data collection and integration techniques using data science are of utmost importance. Leveraging existing data repositories on cancer treatments is crucial for making data available. It is recommended that computer programs be set up to automatically get new data and add it right away to deep neural networks’ search strategies and model selection processes (Zamani et al., 2020). Data from drug response experiments holds immense value, as these experiments involve costly laboratory procedures and the collective efforts of numerous specialists. Hence, researchers must focus on information retrieval techniques that facilitate the availability of newly generated data for modeling purposes.

Transfer learning offers promising results in drug response prediction. Pre-trained deep learning networks have become widely accessible in various contexts. Researchers may encounter challenges in assessing the information that can be extracted from large databases where these models have been pre-trained. Additionally, effectively applying this learning to cancer datasets presents another challenge. The effectiveness of transfer learning depends on the neural network’s ability to capture knowledge from the original data and successfully employ it in a new task involving a cancer database. For instance, transfer learning models are more likely to succeed when valuable knowledge can be absorbed from the original dataset and transferred to the new context.

Through the application of deep learning models, the methodology used in this work has produced relevant, educational, and worthwhile insights into the rapidly developing scientific field of medication response prediction analysis. On the other hand, if there are additional relevant articles, potential biases could appear. However, these studies have a limited effect on the validity of our current results.

Conclusion

The integration of AI algorithms is crucial for developing automated systems for cancer data analysis. Traditional AI and ML algorithms rely on manually engineered features, while DL algorithms learn feature representations from raw input data. DL-based models for drug response prediction outperform traditional ML-based models, but researchers must focus on enhancing their generalizability and interpretability. Proper representation design, network training, and multi-objective optimization algorithms can help address this conflicting objective. Interpretability of DL models with multiple layers is another challenge, and future studies should incorporate strategies for interpretable DL models. DL approaches consistently outperform ML methods in DRP prediction, and DL networks offer various architectures under exploration.

The existing literature on drug response prediction presents several key findings, highlighting the computational aspects and performance of different proposals: DL approaches consistently outperform ML methods in DRP prediction.

Various DL architectures, such as autoencoders, long short-term memory neural networks, and convolutional neural networks, are utilized depending on the data source.

DL models have the potential to improve DRP compared to the limited learning capabilities of traditional ML models.

Overfitting can occur due to data quality problems, neural network architecture, or hyperparameter choices, leading to poor performance.

DL networks offer various architectures under exploration, including ensemble deep neural networks, neural architecture search, and transfer learning approaches.

DL methods have proven more efficient than traditional ML algorithms for predicting the effectiveness of single and multi-drug approaches in anticancer treatments. DL can also be used as an alternative way to learn the internal, low-dimensional representations of pre-screening data. This means that during training, there is less need for explicit computations or detailed feature selection methods. DL methods automatically learn features during the training process.

Additional Information and Declarations

Competing Interests

Author Contributions

Data Availability

The authors declare that they have no competing interests.

Wesam Ibrahim Hajim conceived and designed the experiments, performed the experiments, analyzed the data, prepared figures and/or tables, authored or reviewed drafts of the article, and approved the final draft.

Suhaila Zainudin conceived and designed the experiments, analyzed the data, prepared figures and/or tables, authored or reviewed drafts of the article, and approved the final draft.

Kauthar Mohd Daud conceived and designed the experiments, prepared figures and/or tables, authored or reviewed drafts of the article, and approved the final draft.

Khattab Alheeti conceived and designed the experiments, prepared figures and/or tables, authored or reviewed drafts of the article, and approved the final draft.

The following information was supplied regarding data availability:

This is a literature review.

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
