# Peer review of "Optimized models and deep learning methods for drug response prediction in cancer treatments: a review"

_PeerJ Computer Science, doi:10.7717/peerj-cs.1903_

## Round 0.1 · original submission · Major Revisions

Dear authors,

Thank you for your submission. Your article has not been recommended for publication in its current form. However, we do encourage you to address the concerns and criticisms of the reviewers and resubmit your article once you have updated it accordingly.

Best wishes,

**Language Note:** PeerJ staff have identified that the English language needs to be improved. When you prepare your next revision, please either (i) have a colleague who is proficient in English and familiar with the subject matter review your manuscript, or (ii) contact a professional editing service to review your manuscript. PeerJ can provide language editing services - you can contact us at copyediting@peerj.com for pricing (be sure to provide your manuscript number and title). – PeerJ Staff

Reviewer 1 ·

Basic reporting

The authors used clear and unambiguous English throughout the review. The raw data was not shared as this is a review work.

Experimental design

The survey method is consistent and the sources adequately cited.

The review was organized logically into coherent paragraphs.

The survey methodology was consistent, irrespective of this fact I suggest that the authors bring better understanding of the strength and weaknesses of the used ML and DL algorithms in the reviewed work (line 64-68, 81- 82).

Validity of the findings

The abstract should be reviewed to clearly given the background of the work, the identified gap, the method of used, the comprehensive outcome of the review. The type of the research work should be clearly stated in this section.

The subject was introduced and discuss in detail how Drug Response Prediction (DRP) is critical in various disciplines line 57 needs to be buttressed further .

Table 2 summarized DL based DRP methods in literature. I suggest that an additional column of the algorithms used in modelling in each reviewed work be mentioned as a means to further understand the results from literature. (column 1, column 2, column 3 - algorithm...)

The conclusion identified unresolved questions and further direction.

Reviewer 2 ·

Basic reporting

The article examines recent research that has used several DL algorithms for DRP in cancer therapies. The review analyses current methodologies, discusses procedures, performance, and limits, and seeks to extract knowledge about these algorithms for future investigations. It also examines methods other than supervised and unsupervised learning and provides examples of research that use DL models to predict drug responses in cancer treatment.
The authors still need to effect these comments to increase the quality of their study
• The abstract needs to be written, it looks like an introduction in its present state. The abstract should involve a brief summary of the review paper, including the research question, methodology, main findings, and conclusions.
• The in-text citation is wrong the way it is (line 36, 40). It should be written as thus. For instance, line 40 citation should be written as ((D’Orazio et al., 2022; Tansey et al., 2022; Chang et al., 2018) etc.
• Line 50-51, the authors should try as much as possible to maintain not more than 4 or max 5 authors in-text cited in a sentence.
• The research issue and objectives should be highlighted in a separate paragraph in the introduction section (second to the last paragraph)
• The validity of the study should be comprehensively discussed in a section.
• The Research Questions (RQs) should be formulated for the study and the discoveries should tends to the RQs
• The authors should summarize the remaining part of the article in the last paragraph.
• Authors are advised to comprehensively evaluate relevant literature, identify any shortcomings or gaps in the existing studies, and elucidate how their research addresses and enhances the constraints specified in the literature.
• The authors should add a section called methodology which should explain the criteria used for selecting the literature such as specific databases, keywords, and inclusion and exclusion criteria. They should also describe the search process, any limitations encountered, and the approach to analyzing and synthesizing the literature.
• Line 681, the initials of each of the authors should be removed because in-text citation involves only the authors' surnames and year such as Baymurzina, Golikov & Burtsev, 2022 etc.
• Also et al., in in-text citations should be italicized
• Line 825, change the statement concluding remark to “Conclusion”
• The authors should also read, cite and reference these recent related articles
a. Song, X., Li, Q., & Zhang, J. (2023). A double-edged sword: DLG5 in diseases. Biomedicine & Pharmacotherapy, 162, 114611. doi: https://doi.org/10.1016/j.biopha.2023.114611
b. Li, L., Wang, S., & Zhou, W. (2023). Balance Cell Apoptosis and Pyroptosis of Caspase-3-Activating Chemotherapy for Better Antitumor Therapy. Cancers, 15(1), 26. doi: 10.3390/cancers15010026
c. Zhang, Q., Wang, Y., Bai, R. T., Lian, B. R., Zhang, Y., & Cao, L. M. (2023). X-linked Charcot-Marie-Tooth disease after SARS-CoV-2 vaccination mimicked stroke-like episodes: A case report. World journal of clinical cases, 11(2), 464–471. https://doi.org/10.12998/wjcc.v11.i2.464
d. Zhou, J., Guo, T., Zhou, L., Bao, M., Wang, L., Zhou, W.,... Guo, Z. (2023). The ferroptosis signature predicts the prognosis and immune microenvironment of nasopharyngeal carcinoma. Scientific Reports, 13(1), 1861. doi: 10.1038/s41598-023-28897-2
e. Lu, S., Yang, B., Xiao, Y., Liu, S., Liu, M., Yin, L.,... Zheng, W. (2023). Iterative reconstruction of low-dose CT based on differential sparse. Biomedical Signal Processing and Control, 79, 104204. doi: https://doi.org/10.1016/j.bspc.2022.104204
f. Lu, S., Liu, S., Hou, P., Yang, B., Liu, M., Yin, L.,... Zheng, W. (2023). Soft Tissue Feature Tracking Based on Deep Matching Network. Computer Modeling in Engineering & Sciences, 136(1), 363-379. doi: 10.32604/cmes.2023.025217
g. Dang, W., Xiang, L., Liu, S., Yang, B., Liu, M., Yin, Z.,... Zheng, W. (2023). A Feature Matching Method based on the Convolutional Neural Network. Journal of Imaging Science and Technology. doi: 10.2352/J.ImagingSci.Technol.2023.67.3.030402
h. Liu, M., Zhang, X., Yang, B., Yin, Z., Liu, S., Yin, L.,... Zheng, W. (2023). Three-Dimensional Modeling of Heart Soft Tissue Motion. Applied Sciences, 13(4). doi: 10.3390/app13042493
i. Wang, N., Chen, J., Chen, W., Shi, Z., Yang, H., Liu, P.,... Li, X. (2022). The effectiveness of case management for cancer patients: an umbrella review. BMC Health Services Research, 22(1), 1247. doi: 10.1186/s12913-022-08610-1
j. Zhuang, Y., Chen, S., Jiang, N., & Hu, H. (2022). An Effective WSSENet-Based Similarity Retrieval Method of Large Lung CT Image Databases. KSII Transactions on Internet & Information Systems, 16(7). doi: 10.3837/tiis.2022.07.013
k. Zhuang, Y., Jiang, N., Xu, Y., Xiangjie, K., & Kong, X. (2022). Progressive Distributed and Parallel Similarity Retrieval of Large CT Image Sequences in Mobile Telemedicine Networks. Wireless communications and mobile computing, 2022. doi: 10.1155/2022/6458350
l. Zhang, Z., Wang, L., Zheng, W., Yin, L., Hu, R.,... Yang, B. (2022). Endoscope image mosaic based on pyramid ORB. Biomedical signal processing and control, 71, 103261. doi: 10.1016/j.bspc.2021.103261

Experimental design

The authors should add a section called methodology, which should explain the criteria used for selecting the literature, such as specific databases, keywords, and inclusion and exclusion criteria. They should also describe the search process, any limitations encountered, and the approach to analyzing and synthesizing the literature.

Validity of the findings

The idea of validity in a study pertains to the degree to which the research effectively captures or represents the concepts or phenomena it purports to evaluate. It also considers the extent to which the study effectively captures the intended construct.

Hence, the authors should discuss the validity of their review

---

## Round 0.2 · Minor Revisions

Dear authors,

The reviewers have now commented on your revised manuscript. Although one reviewer is satisfied with the additions and changes, Reviewer 2 writes that his/her comments have not been fully addressed. Please consider the advice and comments of Reviewer 1, especially on clarifying the methodology for study selection, elaborating on data analysis methods, and expanding the discussion section for deeper interpretation and implications, and readability. Furthermore, Reviewer 2 has requested that you cite specific references. You are welcome to add it/them if you believe they are relevant and if you do not include them, this will not influence my decision.

Best wishes,

Reviewer 1 ·

Basic reporting

The review is broad and within the scope of the journal
The introduction adequately address the subject

Experimental design

The methodology is consistent
Adequate sources cited

Validity of the findings

conclusion identified unresolved question

Reviewer 2 ·

Basic reporting

After reviewing the revised article, it appears that the revisions made by the authors were not entirely satisfactory. It is evident that although some of my previous comments were addressed, significant areas still require improvement to enhance the quality of the study. My comments aim to provide constructive feedback to refine the article further, ensuring it meets the high standards expected for publication in this reputable journal. These suggestions are intended to guide the authors in making the necessary adjustments to make the study more robust and acceptable for publication.
1. Ensure the manuscript is structured to clearly outline the objectives, methodology, findings, and conclusions. Each section should smoothly transition into the next, providing a logical flow of information.
2. In the first comments made to the authors, I suggested you go in search of recent articles to be cited and referenced in this article, but I can't seem to find any added upon reviewing the study. It has been noted that the content appears outdated, mainly due to the lack of inclusion of recent literature from 2023. It's crucial to reference and discuss the latest findings and advancements for a study in such a rapidly evolving field. I strongly recommend that the authors conduct an updated literature review, focusing on recent articles from 2023 and 2024, and integrate these findings into the paper, especially in the introduction and literature review sections. This inclusion will not only enhance the relevance and timeliness of the study but also ensure that it reflects the current state of research in the field, making it a more valuable contribution to this reputable journal.
Citing recent articles is crucial for maintaining the integrity, relevance, and progression of academic research. It benefits both the authors in terms of credibility and advancement of their work and the journals in terms of visibility and impact. Hence, I have suggested publications from 2023 and 2024 relating to your work for you to read, cite, and reference in your article introduction and literature review section.
a. Yan, J., Liu, D., Wang, J., You, W., Yang, W., Yan, S.,... He, W. (2024). Rewiring chaperone-mediated autophagy in cancer by a prion-like chemical inducer of proximity to counteract adaptive immune resistance. Drug Resistance Updates, 73, 101037. doi: https://doi.org/10.1016/j.drup.2023.101037
b. Wang, H., Yang, T., Wu, J., Chen, D., & Wang, W. (2023). Unveiling the Mystery of SUMO-activating enzyme subunit 1: A Groundbreaking Biomarker in the Early Detection and Advancement of Hepatocellular Carcinoma. Transplantation Proceedings, 55(4), 945-951. doi: https://doi.org/10.1016/j.transproceed.2023.03.042
c. He, B., Sun, H., Bao, M., Li, H., He, J., Tian, G.,... Wang, B. (2023). A cross-cohort computational framework to trace tumor tissue-of-origin based on RNA sequencing. Scientific Reports, 13(1), 15356. doi: 10.1038/s41598-023-42465-8
d. Xu, H., Li, L., Wang, S., Wang, Z., Qu, L., Wang, C.,... Xu, K. (2023). Royal jelly acid suppresses hepatocellular carcinoma tumorigenicity by inhibiting H3 histone lactylation at H3K9la and H3K14la sites. Phytomedicine, 154940. doi: https://doi.org/10.1016/j.phymed.2023.154940
e. Zhao, J., Liu, Y., Zhu, L., Li, J., Liu, Y., Luo, J.,... Chen, D. (2023). Tumor cell membrane-coated continuous electrochemical sensor for GLUT1 inhibitor screening. Journal of Pharmaceutical Analysis, 13(6), 673-682. doi: https://doi.org/10.1016/j.jpha.2023.04.015
f. Chen, L., He, Y., Zhu, J., Zhao, S., Qi, S., Chen, X.,... Xie, T. (2023). The roles and mechanism of m6A RNA methylation regulators in cancer immunity. Biomedicine & Pharmacotherapy, 163, 114839. doi: https://doi.org/10.1016/j.biopha.2023.114839
g. Shen, W., Pei, P., Zhang, C., Li, J., Han, X., Liu, T.,... Yang, K. (2023). A Polymeric Hydrogel to Eliminate Programmed Death-Ligand 1 for Enhanced Tumor Radio-Immunotherapy. ACS Nano. doi: 10.1021/acsnano.3c08875.
3. The methodology section should clearly explain the criteria for selecting the studies included in the review. This consists of the databases searched, the keywords used, and the time frame of the literature considered.
4. Provide more details on how the data from the selected studies were analyzed. These should be explained clearly if any quantitative analysis or meta-analysis methods were used.
5. Expand the discussion section to interpret the findings more deeply. How do these findings contribute to the existing body of knowledge? What are the implications for future research, and what gaps in knowledge remain?
6. It has been observed that the manuscript contains numerous grammatical errors, which significantly impact its readability and professional presentation. To address this issue, I recommend that the authors seek the assistance of an English language expert or a professional editor. This step is vital to ensure the manuscript meets the linguistic standards required for publication in an esteemed journal.

Experimental design

The methodology, relying on a word cloud from titles and abstracts, may introduce selection bias and not fully represent the diversity of deep learning techniques. Challenges noted include the generalization of models to new data types, the 'black box' nature of deep learning leading to interpretability issues, and the need for more effective ensemble methods to enhance model robustness. There is an emphasis on the need for strategies to minimize these biases in clinical applications. This underscores the necessity for more inclusive research approaches, as well as advancements in model transparency and adaptability, to fully harness the potential of deep learning in medical diagnostics and treatment planning.

Validity of the findings

No comment

Additional comments

Based on my observation, there are a few grammatical and punctuation errors, suggesting a need for significant language editing. In such cases, it's advisable for the authors to seek assistance from an English language expert or a professional editor. This expert can help refine the language, ensuring that the text is clear, coherent, and professionally presented. Accurate and polished language is crucial, especially in academic and scientific writing, as it affects the readability and credibility of the work. Engaging with a language expert will not only correct grammatical and punctuation errors but also enhance the overall flow and comprehension of the document, making it more accessible to its intended audience.

---

## Round 0.3 · accepted · Accept

Dear authors,

Thank you for the revision and for clearly addressing all the reviewers' comments. I confirm that the paper is improved and addresses the concerns of the reviewers. Your paper is now acceptable for publication in light of this revision.

Best wishes,

Reviewer 2 ·

Basic reporting

After thoroughly reviewing your manuscript's content, methodology, and overall contribution, I am pleased to recommend its acceptance for publication. The authors have presented a well-structured, rigorously researched, and eloquently articulated work that significantly contributes to the research domain.

Experimental design

No comment

Validity of the findings

No Comment

Additional comments

No Comment